# Sixteen-Year Follow-Up in a Cavernous Sinus Hemangiopericytoma: Improved Outcomes over Radiotherapy Advances

**DOI:** 10.3390/brainsci12091209

**Published:** 2022-09-08

**Authors:** Beatrice Detti, Lilia Bardoscia, Antonio Rosario Pisani, Salvatore Cozzi, Manuele Roghi, Paolo Mammucci, Angela Sardaro

**Affiliations:** 1Radiotherapy Unit, Azienda Ospedaliera-Universitaria Careggi, 50134 Florence, Italy; 2Radiation Oncology Unit, S. Luca Hospital, Healthcare Company Tuscany Nord Ovest, 55100 Lucca, Italy; 3Nuclear Medicine Unit, Interdisciplinary Department of Medicine, University of Bari Aldo Moro, 70125 Bari, Italy; 4Radiation Therapy Unit, Azienda USL-IRCCS di Reggio Emilia, 42123 Reggio Emilia, Italy; 5Department of Interdisciplinary Medicine, Section of Diagnostic Imaging and Radiotherapy, University Aldo Moro, 70125 Bari, Italy

**Keywords:** intracranial hemangiopericytomas, surgery, radiotherapy, stereotactic radiosurgery, Gamma Knife

## Abstract

Intracranial hemangiopericytomas are rare tumors, accounting for 1% of all central nervous system malignancies. This tumor is considered at high risk of local and also distant metastases. Surgical excision is the gold standard for treatment, but it is seldom curative by itself. Adjuvant radiotherapy is often recommended. We report an overview and update of the available literature on one such rare but aggressive mesenchymal tumor, using the case of a 46-year-old woman affected by hemangiopericytoma of the cavernous sinus surgically removed and treated with adjuvant radiotherapy at our institution. After seven years, the patient underwent a local recurrence and was treated with exeresis and Gamma Knife radiotherapy. Sixteen years after the initial diagnosis, she is still well with stable disease.

## 1. Introduction

Hemangiopericytoma (HPC) is a rare malignancy that develops from Zimmerman’s pericytes, contractile spindle cells located around capillaries and post-capillary venules [1]. Clinical and radiological characteristics of this kind of tumor often make them indistinguishable from meningiomas and often contribute to an early misclassification as an angioblastic variant of meningioma. The lower extremities, retroperitoneum, or pelvis, head, and neck are the most frequent locations of HPC, although this tumor has also been described in other extracranial sites [2,3]. Intracranial location of HPC is rare, accounting for 1% of all central nervous system (CNS) tumors and 2–4% of meningeal tumors [4,5,6]. Intracranial HPCs almost always present as solitary, supratentorial, dural-based lesions, often arising from the falx, tentorium, dural sinuses, and/or the skull base. Intracranial HPC is slightly predominant in males, and the peak incidence is in the fourth and fifth decades of life. Intracranial HPC has been considered a low-grade malignancy (WHO grade II) in the past, with anaplastic variants classified as grade III [7]. In 2016, the World Health Organization (WHO) classification of CNS tumors, solitary fibrous tumor (SFT) and HPC, were restructured as one entity (known as SFT/HPC). A new grading system, different from the typical WHO CNS tradition, was also proposed for low-grade SFT (WHO grade I) and higher-grade lesions HPC (WHO grade II) and anaplastic HPC (WHO grade III) [8]. The 2021 WHO update removed the term “hemangiopericytoma” with the aim of conforming to the WHO non-CNS tumor nomenclature, while the CNS modified option of three grades was retained [9]. Higher-grade SFT tumors have been shown to have worse survival outcomes [10]. The mTORC1 signaling pathway has been supposed to be involved in HPC tumorigenesis from pericytes in animal models, possibly sharing a common molecular pattern with the STAT6 axis, which has been reported to drive the development of human HPC [11].

The aggressive biology confers to this neoplasm a significant risk of local relapse and distant metastases, mainly to bone, lung, and liver [12]. Guthrie et al. reported a 5-year, 10-year, and 15-year incidence of metastases of 13%, 33%, and 64%, respectively [13]. The radiological diagnosis of HPC is often difficult, mainly because the intracranial location is similar to meningioma [6,13]. Intracranial HPC typically appears as a hyper- or isodense lesion on a CT scan, with homogeneous contrast enhancement without calcifications [14]. Magnetic Resonance Imaging (MRI) appearance of such lesions consists of an isointense area on T1 and T2-weighted sequences, often heterogeneous [15,16]. This neoplasm usually shows a typical angiographic aspect, with possible blood supply from both external or internal carotid arteries and a high number of small vessels within the tumor mass [17]. A definitive diagnosis is often made on histological specimens. The main criteria are the presence of many capillary blood vessels, cells with chromatin-rich nuclei surrounded by pericytes, and a typical “staghorn-like” vascular pattern with lots of branching vessels of variable wall thickness [18,19,20]. Surgical excision is the gold standard for the treatment of HPC, but adjuvant radiotherapy is often recommended to improve the recurrence-free interval [21]. Recent reports also support the use of stereotactic radiosurgery for the treatment of residual or recurrent intracranial HPC to improve local control of the disease [22].

The purpose of this paper is to describe the clinical and radiological features and the results of treatment in a patient with intracranial HPC, treated with postoperative radiotherapy in 2005, stereotactic re-irradiation for in-field tumor recurrence in 2012, and still negative at the last follow-up visit.

## 2. Case Presentation

In December 2005, a 46-year-old woman underwent a brain MRI after complaining of a headache for 4 months, revealing a tumor mass in the right temporal lobe (Figure 1).

The tumor lesion was close to the lateral wall of the cavernous sinus and appeared as an extra-axial lesion with inhomogeneous contrast enhancement due to a central necrotic area and a peripheral solid area with perilesional edema. At the subsequent neurologic examination, the patient felt drowsy, with response to the verbal stimuli and deficit of the third right cranial nerve. A right fronto-temporal craniotomy was then performed, and the macroscopic examination showed a hypervascular solid mass in the medial cranial fossa, which was completely removed. The histological exam revealed hemangiopericytoma (WHO grade II). Immunohistochemistry was carried out and showed the following parameters: S100, SY, NF, NSE, EMA, GFAP negative; CD34, VIM positive; Mib-1: 7%. The patient achieved the complete disappearance of the previous neurological symptoms within one month after surgery. A postoperative brain MRI was performed 40 days after surgery, with no evidence of any macroscopic residual disease. Staging chest and abdomen CT scans did not show any evidence of extracranial HPC, as well as spinal MRI. Five weeks after surgical excision, the patient received adjuvant radiotherapy, for a total dose of 5400 cGy, in 30 daily fractions of 180 cGy. CT simulation was performed with 3-mm-wide slices. The preoperative brain MRI was co-registered with planning CT images on a Pinnacle treatment planning system (TPS), version 8 (Philips Medical Systems, Andover, MA, USA). The clinical target volume (CTV) was defined on the preoperative MRI as the tumor lesion and the perilesional edema, plus an isotropic wide margin of 1 cm. A 5-mm isotropic expansion of the CTV was applied to create the planning target volume (PTV). The total PTV size was 129.32 cc. The maximum dose for PTV coverage was 5632 cGy, the minimum dose was 3948 cGy, and the mean dose was 5348 cGy. The treatment was administered with 6-MV photons. A linear accelerator (Elekta, Synergy) and three non-coplanar fields were used. The treatment was well tolerated, and there was no need for steroid supplement during treatment delivery. At the end of the treatment, the patient was enrolled in our protocol of follow-up, consisting of quarterly brain MRI and neurologic examination for the first two years and every six months thereafter. At the follow-up visit performed in November 2011, the patient was alive and well, without any radiological evidence of recurrence (Figure 2).

In June 2012, a new brain MRI revealed a suspected in-field disease relapse in the right cavernous sinus. Clinically, the patient did not complain of any neurological symptoms; the restaging chest-abdomen CT scan was negative.

The clinical case was discussed in a multidisciplinary neuro-oncologic tumor board, and the patient underwent surgical excision of the recurrent lesion. Histological examination confirmed hemangiopericytoma, WHO grade III. Immunohistochemistry showed the following parameters: EMA negative; CD34 positive; Mib-1 22%.

The postoperative MRI showed a residual pathological nodule adherent to the lateral portion of the right cavernous sinus. After further multidisciplinary evaluation of the clinical case, the previous radiation treatment plan was unarchived, and we decided to re-treat the patient with Gamma Knife fractionated stereotactic radiotherapy, for a total dose to the target volume of 2400 cGy, in 4 fractions of 600 cGy. The treatment was well tolerated; the patient did not complain of any symptoms. At the last follow-up, performed in December 2021, the patient was alive and well; the brain MRI showed a stable reduction of the residual disease in the right cavernous sinus. (Figure 3).

## 3. Discussion

The histopathogenesis of HPC has been controversial for a long time. In 1942, Stout and Murray identified for the first time an extracranial soft tissue tumor consisting of proliferating pericytes and called it “hemangiopericytoma” [18]. In 1954, Begg and Gart reported the first case of a cranial HPC [23]. In the 1960s, in-vivo microscopic observation of transplantable HPC in hamster models was first described as dense agglomerations of tumor cells and fibrovascular spaces containing capillaries and bundles of collagen fiber, some of them collapsed, initially with an arboreal aspect, then acquiring a lobulated vascular pattern with the tumor growth [24].

Although this tumor arises from meningeal capillary pericytes, it seems clear that HPC behaves differently than meningioma. They both have the same high tendency to local recurrence, but even when local control can be achieved, the risk of distant metastases remains for HPC. The identification of the NAB2-STAT6 gene fusion in both SFT and HPC led to the reclassification of low-grade SFT and higher-grade lesions HPC and anaplastic HPC as one entity in the 2016 WHO classification of CNS tumors [8]. HPC could be classified as a WHO grade II or III tumor, with anaplastic features like a mitotic index ≥ 5 per 10 high power fields (HPF) and/or necrosis, plus at least two of the following: hemorrhage, moderate to high nuclear atypia, and cellularity [7,8]. However, the updated SFT/HPC WHO grading system may have introduced new challenges in assessing the malignancy risk of such rare but aggressive meningeal tumors [25]. The term “hemangiopericytoma” was retired in the 2021 WHO update to conform to the WHO non-CNS soft tissue tumor classification, although the proper CNS-modified 3-grading scheme was still adopted [9]. Recently, mTORC1 has been supposed to be a driver of HPC in murine models. Loss of the tumor suppressor gene Tsc2 in pericytes gave rise to the formation of HPC in multiple selected sites (not in lung or kidney) [11]. Such findings suggested that the two genetic modifications (mTORC1 in mice and STAT6 in human HPC) might derive from a common signaling pathway in pericytes, but other cell types and/or the microenvironment are also likely to affect tumor development [11].

The median age of presentation in patients with HPC ranges from 38 to 42 years in different series, which is lower than the median age of meningiomas [13]. Due to radiological characteristics similar to meningioma, the radiological diagnosis of such unusual, high-grade SFT is often difficult [6,13]. On MR imaging, HPC lesions usually appear as heterogeneous, T1 and T2-weighted isointense areas [16]. Advanced, functional MR sequences such as diffusion-weighted imaging (DWI) may provide additional information on tumor microstructural characteristics. In fact, SFT/HPC have shown higher apparent diffusion coefficient (ADC) values than meningiomas [26]. In recent years, machine learning–based radiomics analysis, combining MRI-based radiomic features and clinical findings, has been proposed as a viable tool to make a differential diagnosis of HPC and meningioma with high accuracy [27,28].

We presented the case of a 46 years-old woman affected by intracranial hemangiopericytoma surgically removed and treated with adjuvant radiotherapy at our institution. Such a primary, multimodal therapeutic approach provided local control of disease and a disease-free interval of 7 years. Upfront radical surgery is the treatment of choice for newly diagnosed HPCs [29]. Because of their tendency to recur, resection must be radical whenever possible. Actually, there are no validated criteria for surgical resection [30]. The extent of resection is reported in many studies as an important factor influencing local control; more controversial is the impact surgery may have on overall survival. Two published series by Soyuer and Kim showed better local control in patients treated with gross-total removal than patients who underwent partial excision, but they were not able to find a statistically significant effect on overall survival [31,32]. On the other hand, Guthrie et al. reported a similar time to first recurrence in both patients treated with radical or partial removal of the lesion, but they found a statistically significant impact of surgery on overall survival (109 vs. 65 months) [13]. In another retrospective study in 2011, Schiariti et al. found that gross-total resection followed by adjuvant external-beam radiotherapy (EBRT) provided patients with the highest probability of an increased recurrence-free interval and overall survival [33]. Additionally, Ghia et al., in one of the largest single series of patients with intracranial HPC reported in the literature (number of patients = 88), showed a strong correlation between the extent of resection and both cause-specific survival and overall survival on multivariate analysis [34]. Since the invasive nature of HPC and the involvement of dural sinuses may often preclude a radical excision, the use of postoperative radiotherapy (RT) seems to be attractive. Ciliberti and colleagues investigated the role of radiation in the management of such an unusual mesenchymal tumor [35]. Currently, a multimodal management—surgery followed by adjuvant RT—is recommended to treat localized disease [36]. However, the results reported in the literature are controversial. Adjuvant RT, either EBRT or Gamma Knife stereotactic radiosurgery (SRS) and/or proton beam therapy, has proven to increase the recurrence-free interval but does not have a clear impact on overall survival (OS) [37,38]. As stated before, one of the largest retrospective series supporting the efficacy of improving local control of radiation therapy was published by Guthrie and Dufour. Among the 44 HPC cases reported by the former, those receiving adjuvant RT had a significantly increased disease-free survival time (mean of 74 vs. 29 months, *p* < 0.05), as well as a longer OS (92 vs. 62 months) [13]. Moreover, the authors defined a dose-response relationship without local recurrences among patients receiving ≥50 Gy. Dufour et al. recorded 17 patients with HPC who were followed for 34 years. The local recurrence rate was 12.5% in patients treated with postoperative RT, compared to 88% after surgery alone [39]. In 2001, Alen et al. reported eight intracranial HPC patients treated with EBRT at some time during the course of their disease; among these, five patients were treated after surgery and received doses ranging from 60 to 64 Gy. Four of them survived 6, 15, 16, and 85 months without evidence of recurrence, respectively, and the remaining one, who had only subtotal resection, died 76 months after surgery [14]. A recent series of 48 patients by Haas and colleagues showed 90% of local control after surgery and adjuvant RT, compared to 60% after surgical excision alone (*p*: 0.052), but local control probability was significantly associated with adjuvant RT (HR 0.25; *p* ≤ 0.001), while there was no impact of multimodal treatment on OS [40]. In 2004, Soyuer et al. reported the M. D. Anderson Cancer Center experience over a 20-year period; surgery was the primary treatment for all 29 patients. Adjuvant RT was performed in 10 cases, with delivered doses of 3340–6120 cGy (median, 5400 cGy). Postoperative irradiation did not have a significant effect on local control after radical exeresis in this case (*p* = 0.18), but it could be attributable to the limited number of patients [31]. In this study, as stated by the authors, it is noteworthy that none of the 3 patients undergoing adjuvant RT after gross-total removal of the lesion locally relapsed, whereas 5 of 11 patients treated with radical surgery and radiation therapy experienced local recurrence. Additionally, Kim et al. did not find adjuvant RT to have a statistically significant effect on local control of disease. However, the 5-year recurrence-free survival (RFS) rates associated with complete excision followed by postoperative RT were higher than radical surgery alone (100% and 70%, respectively) [32]. Rutkowski et al. published another series of 40 consecutive patients with intracranial HPC [41]. Radiotherapy was performed in 9 of 16 patients who underwent radical surgery and in 14 of 19 patients with subtotal removal of the lesion. The authors did not demonstrate a statistically significant benefit in terms of time to recurrence with the prescription of adjuvant RT. Despite this, they showed a trend towards statistical significance, with an improvement of median time to recurrence from 3.9 years in patients not treated with RT to 6.6 years in the group who underwent RT (*p* = 0.082). A retrospective study of 15 patients by Kumar et al. showed that postoperative RT (median dose 50 Gy) led to a median time to local recurrence of 68 months vs. 37 months in patients who did not receive it, but also that adjuvant treatment did not confer any significant protection against the development of distant metastases [42]. Jeon and colleagues also reported a decrease in local tumor recurrence with the addition of RT after surgery but no differences in OS or distant metastases-free survival (DMFS) [43]. A retrospective comparison between 29 patients treated with adjuvant EBRT (intensity-modulated radiotherapy (IMRT) or SRS) and 37 patients undergoing upfront surgery alone showed no impact of postoperative radiation on local tumor control nor survival, but similar local control rates after IMRT and SRS, therefore, they both remain viable therapeutic options [44]. In the retrospective analysis of Ghia et al., there was a strong correlation between the use of upfront postoperative irradiation and improved survival outcome. The authors recommend an RT dose ≥ 60 Gy to optimize local control of disease [34]. None of these studies were able to definitely demonstrate the advantage of adjuvant RT, especially in the group of completely resected patients. The main limit of these studies is obviously represented by the small number of involved patients, which means a small statistical power. Nevertheless, almost all authors’ conclusions favor the use of adjuvant RT, at least in patients treated with no radical surgery. A Korean multicenter study by Lee et al. recently confirmed the role of postoperative radiation in improving disease control of intracranial HPC. Among 133 included patients, they found 10-year progression-free survival (PFS) and OS rates of 45% and 71%, respectively. Moreover, there were significantly higher local control rates and PFS than surgery alone, irrespective of the surgical extent and WHO grading (*p* < 0.001 in both cases), as long as to enclose the tumor bed with a large, 1.0–2.0 cm margin of the target volume [45]. It is also known that hemangiopericytoma tends to relapse also after a gross-total removal of the lesion. For this reason, in our opinion, limited field radiotherapy should also be prescribed to this group of patients. Another controversial issue is represented by the optimal radiation total dose to achieve good control of the disease. It has been shown that receiving more than 45 Gy is necessary to have fewer local disease recurrences. Guthrie et al. reported that seven of eight patients receiving less than 45 Gy irradiation experienced local disease relapse, while only two of nine patients treated with more than 45 Gy on the tumor bed locally relapsed. None of the three patients who received more than 50 Gy experienced disease recurrence [13].

Dufour et al. reported similar results; they recommended treating HPC patients with more than 50 Gy [39]. In the series reported by Straples and colleagues, a dose greater than 55 Gy led to permanent local control of HPC [46]. In recent years, there has been an increasing interest in the use of new radiotherapy techniques such as stereotactic radiotherapy (SRT) or radiosurgery (SRS) as salvage treatment for residual or recurrent HPC [36]. Kim et al. found that SRT is an effective and safe adjuvant management option for patients with recurrent or residual HPCs. They reported on nine patients with HPC that underwent SRT with Gamma Knife for the treatment of 17 recurrent or residual lesions. Tumor control was achieved in 14 out of 17 lesions (82.4%), with an actuarial local tumor control rate at 5 years after SRT of 67%. Marginal dose > 17 Gy was the only statistically significant factor for local tumor control on univariate analysis [47]. Payne et al. reported the results of 12 patients with intracranial HPCs treated with Gamma Knife SRS; 8 patients showed a response with a degree of regression ranging from 30–80% [48]. Sheehan and colleagues recommended a radiation dose of at least 15 Gy for successful tumor control of 14 patients treated with radiosurgery. Local tumor control and 5-year survival rates were 76% and 100%, respectively [49]. Kano et al. reported significantly better PFS in 20 HPC patients treated with a high marginal dose (>14 Gy). The OS after radiosurgery was 100%, 86%, and 14% at 1, 5, and 10 years, respectively. Tumor control was reached in 73% of patients [50]. More recent reports, although retrospective, also supported SRT for the treatment of residual or recurrent intracranial HPC [21,51,52]. In particular, a large multicenter study through the International Gamma Knife Research Foundation reviewed the medical records of 90 patients with recurrent HPC treated with Gamma Knife radiosurgery (133 treated lesions in total). A median margin dose of 15 Gy (range 2.8–24 Gy) and a maximum dose of 32 Gy (range 8–51 Gy) was delivered. After a median follow-up of 59 months (range 6–190), a 59.4% local tumor control and low risk of adverse events were reported; 2-year, 4-year, and 10-year PFS were 81.7%, 66.3%, and 25.5%, respectively, while 2-year, 4-year, and 10-year OS were 91.5%, 82.1%, and 53.7%, respectively [53]. In line with the available literature, the presented case underwent local recurrence 7 years after primary treatment and was treated with surgery and a further course of stereotactic radiotherapy. Sixteen years after the initial diagnosis, she was still alive and well with stable disease.

Its aggressive biology often confers the rare but aggressive mesenchymal CNS tumor a high risk of local relapse and also of distant metastases, for which surgical removal is the gold standard for treatment, but it is seldom curative by itself. To support this, adjuvant radiotherapy has been reported to improve the recurrence-free interval, so a multidisciplinary discussion and management of HPC should always be recommended.

## 4. Conclusions

Hemangiopericytoma is an unusual, high-grade malignant tumor with a high recurrence rate after surgery. Unfortunately, the available series are still so low that there is no standard treatment for HPC. Upfront surgical removal, possibly with radical intent, is the treatment of choice to obtain great survival outcomes. To date, a clear impact of postoperative irradiation on overall survival has not been demonstrated. However, adjuvant radiotherapy has been reported to improve the recurrence-free interval, so multimodal management of HPC is actually desirable. Despite controversial results, in our opinion, postoperative radiotherapy should be considered as part of the initial management, both in the case of radical and partial surgical excision, to improve local control of disease. In this setting, the radiation dose should be at least 50 Gy. Stereotactic radiotherapy or radiosurgery may be favorably considered also for the treatment of residual tumors or local relapse. Further studies are needed to verify if high radiotherapy doses could improve the outcomes.

## Figures and Tables

**Figure 1 brainsci-12-01209-f001:**
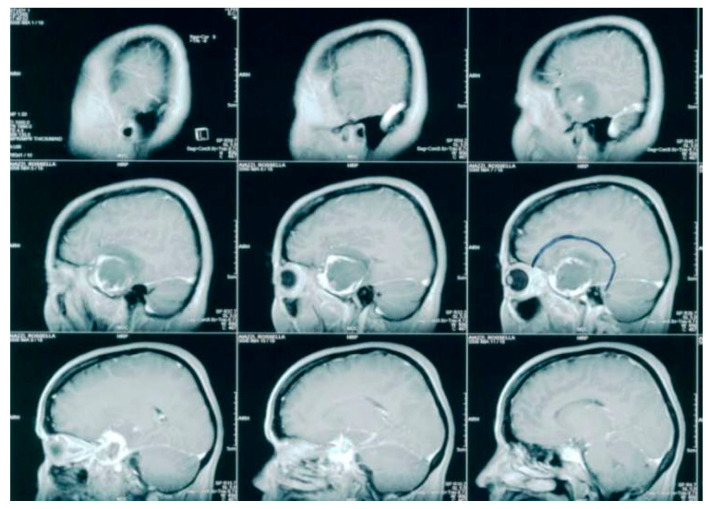
Early sagittal contrast-enhanced T1-weighted sequence images MRI (performed at the initial diagnosis) demonstrated a spherical, extra-axial lesion in the right temporal region. The tumor presented with a characteristic peripheral enhancement that could suggest the hypothesis of hemangioperycitoma.

**Figure 2 brainsci-12-01209-f002:**
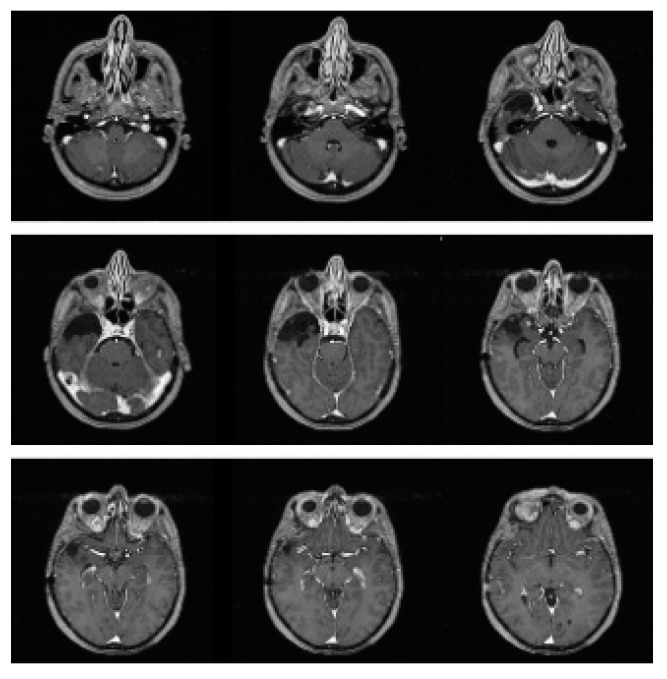
Contrast-enhanced T1-weighted sequence images MRI (performed after surgery and radiotherapy) showed radical removal of the lesion, residual malacic area with no signs of relapsing disease, and no pathological enhancement.

**Figure 3 brainsci-12-01209-f003:**
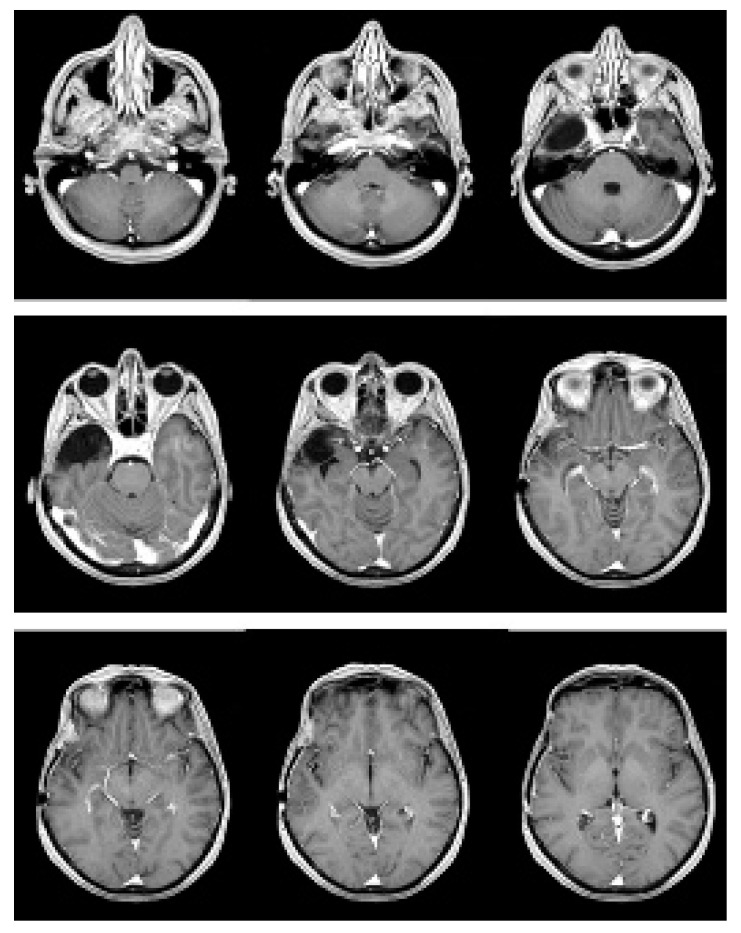
Contrast-enhanced T1-weighted sequence images MRI, performed after partial removal of the recurrence and Gamma Knife.

## Data Availability

Not applicable.

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
