# Peer review of "Sixteen-Year Follow-Up in a Cavernous Sinus Hemangiopericytoma: Improved Outcomes over Radiotherapy Advances"

_brainsci, 2022, doi:10.3390/brainsci12091209_

Round 1

Reviewer 1 Report

Some points need to be addressed, look at these points:

- Title: "cavernous sinus hemangiopericytoma and review of the literature", but paper is just a case report. Please provide a diagram flow of this review according with PRISMA guidelines and add a literature review section in Results.

- Authors reported the old 2016 WHO classification. Please report the new one: Louis et al. The 2021 WHO Classification of Tumors of the Central Nervous System: a summary. Neuro Oncol. 2021 Aug 2;23(8):1231-1251. doi: 10.1093/neuonc/noab106. 

- Conclusion section. What does this paper add new to the literature?

- Figures 1-3 are of low quality and need improvement.

Author Response

- Title: "cavernous sinus hemangiopericytoma and review of the literature", but paper is just a case report. Please provide a diagram flow of this review according with PRISMA guidelines and add a literature review section in Results.

We apologize for the inconvenience. We did not perform a systematic review of the literature. We aimed to provide an overview of the available literature and contextualize our promising findings.

The title was revised and updated.

- Authors reported the old 2016 WHO classification. Please report the new one: Louis et al. The 2021 WHO Classification of Tumors of the Central Nervous System: a summary. Neuro Oncol. 2021 Aug 2;23(8):1231-1251. doi: 10.1093/neuonc/noab106.

We apologize for the inconvenience.

The text was revised and updated.

- Conclusion section. What does this paper add new to the literature?

Thank you for your suggestions.

Our aim was to arouse interest about and support a safe and effective therapeutic option as radiotherapy might be for unusual CNS tumors, as the available series are still so low that there is not a standard for treatment.

We believe that our case report can be used for further research in the field.

We also provided an overview and update the available literature on such a rare, but aggressive mesenchymal tumor, with a focus on such unusual point of view: the role of radiation treatment, in the setting of primary treatment at onset, but also for intracranial reirradiation, especially given the advent of stereotactic radiation techniques.

The manuscript was revised and updated.

- Figures 1-3 are of low quality and need improvement.

We apologize for the inconvenience.

The quality of Figures 2 and 3 was improved.

Unfortunately, Figure 1 could not be further improved, due to the long time interval since that images were acquired (2005), and major difficulties in obtaining healthcare permission to access and unarchive so outdated DICOM files.

Reviewer 2 Report

The manuscript presented from Detti et al., is interesting and original, it is a case report that can be used for further research in the field. However I would suggest the author to improve introduction and discussion including preclinical studies (previous) that used animal models to study this pathology. This will improve the interest of readers.

Author Response

The manuscript presented from Detti et al., is interesting and original, it is a case report that can be used for further research in the field. However I would suggest the author to improve introduction and discussion including preclinical studies (previous) that used animal models to study this pathology. This will improve the interest of readers.

Thank you for your appreciation and suggestion.

The text has been revised and updated.

Round 2

Reviewer 1 Report

Good

Reviewer 2 Report

The authors improved the quality of the manuscript. I would propose the acceptance in the present form.